# The Antagonism of the Prokineticin System Counteracts Bortezomib Induced Side Effects: Focus on Mood Alterations

**DOI:** 10.3390/ijms221910256

**Published:** 2021-09-23

**Authors:** Giada Amodeo, Benedetta Verduci, Patrizia Sartori, Patrizia Procacci, Vincenzo Conte, Gianfranco Balboni, Paola Sacerdote, Silvia Franchi

**Affiliations:** 1Department of Pharmacological and Biomolecular Sciences, Università degli Studi di Milano, Via Vanvitelli 32, 20129 Milan, Italy; giada.amodeo@unimi.it (G.A.); benedetta.verduci@studenti.unimi.it (B.V.); paola.sacerdote@unimi.it (P.S.); 2Department of Biomedical Sciences for Health, Università degli Studi di Milano, Via Colombo 71, 20133 Milan, Italy; patrizia.sartori@unimi.it (P.S.); patrizia.procacci@unimi.it (P.P.); vincenzo.conte@unimi.it (V.C.); 3Department of Life and Environmental Sciences, University of Cagliari, Via Ospedale 72, 09124 Cagliari, Italy; gbalboni@unica.it

**Keywords:** prokineticin system, chemotherapy-induced side effects, neuroinflammation, neuropathic pain, anxiety, depression

## Abstract

The development of neuropathy and of mood alterations is frequent after chemotherapy. These complications, independent from the antitumoral mechanism, are interconnected due to an overlapping in their processing pathways and a common neuroinflammatory condition. This study aims to verify whether in mice the treatment with the proteasome inhibitor bortezomib (BTZ), at a protocol capable of inducing painful neuropathy, is associated with anxiety, depression and supraspinal neuroinflammation. We also verify if the therapeutic treatment with the antagonist of the prokineticin (PK) system PC1, which is known to contrast pain and neuroinflammation, can prevent mood alterations. Mice were treated with BTZ (0.4 mg/kg three times/week for 4 weeks); mechanical allodynia and locomotor activity were evaluated over time while anxiety (dark light and marble burying test), depression (sucrose preference and swimming test) and supraspinal neuroinflammation were checked at the end of the protocol. BTZ treated neuropathic mice develop anxiety and depression. The presence of mood alterations is related to the presence of neuroinflammation and PK system activation in prefrontal cortex, hippocampus and hypothalamus with high levels of PK2 and PKR2 receptor, IL-6 and TNF-α, TLR4 and an upregulation of glial markers. PC1 treatment, counteracting pain, prevented the development of supraspinal inflammation and depression-like behavior in BTZ mice.

## 1. Introduction

The number of cancer survivors is increasing; however, the development of side effects due to chemotherapeutic treatment is frequent. Among these, chemotherapy-induced painful neuropathy (CIPN), chemotherapy-induced cognitive impairment (CICI) and psychological distress such as anxiety and depression deserve particular attention considering their incidence and impact on patient’s life quality [1,2,3,4]. All these complications are independent from the antitumoral effect and are also common to newer chemotherapeutic agents such as proteasome inhibitors [5,6]. Even if the underlying mechanisms are not clear, it is known that chemotherapy-induced side effects have a multifactorial origin often related to oxidative stress and neuroinflammation [4,7,8,9,10]. Moreover, it was suggested that all these complications are commonly interconnected [11,12,13]; it is in fact recognized a close relationship between pain and psychiatric disorders due to a neurobiological overlapping between pain processing, stressful and emotional signals. In addition, current theories suggest that long-term pain triggers neural changes that would be responsible for affective and cognitive alterations such as anxiety and depression [11,12,14,15,16,17]. On the other hand, chemotherapeutics can directly exert a toxic effect in the central nervous system (CNS). Indeed, even if many anticancer drugs cannot cross the blood–brain barrier (BBB), their metabolites may cross it resulting in molecular, structural and functional changes within the CNS. In addition, BBB could also be affected by ROS release and TLR4 activation occurring in brain areas after chemotherapy [7,18,19]. Hippocampus (HPC) and prefrontal cortex (PFC) are key sites in mood control [20,21] as well as important regulators of the affective and emotional component of pain and are also involved in hypothalamus-pituitary-adrenal (HPA) axis regulation [21,22]. Neurochemical and morphological changes in these brain areas have been observed both in chronic pain conditions and in depressed subjects. In particular, an increase in proinflammatory cytokines like IL-6 and TNF-α and a decrease in the neurotrophin BDNF are often associated with anxiety and depression in patients as well as experimental models [20,23,24,25,26,27]. We have recently described a role of a newly discovered chemokine, the prokineticin (PK) 2, in the development of CIPN induced by two different chemotherapeutics: bortezomib (BTZ) and vincristine (VCR). We identified PK2 as a key molecule in the neuroinflammatory pathway that moving from the peripheral (PNS) to central nervous system (CNS) is important for pain processing [28,29]. In addition to being an important regulator of inflammation and pain, PK2 is also involved in mood regulation [30]. Intracerebroventricular injection of PK2 in mice leads to increase in anxiety- and depressive-like behavior while mice deficient in *pk2* gene display reduced anxiety and depression [30].

In this paper, using the proteasome inhibitor bortezomib, at a protocol able to induce pain and to activate a neuroinflammatory cascade that plays a role in the development and maintenance of peripheral neuropathy, we aim to:verify if, in the experimental model of BTZ-induced neuropathic pain, the presence of hypersensitivity is related to the presence of emotional alterations such as anxiety and depression;evaluate the role of neuroinflammation and the involvement of the PK system in such alterations;verify whether the treatment with the PKRs antagonist PC1, at a protocol able to counteract pain, can prevent the development of mood alterations.

## 2. Results

### 2.1. Effect of BTZ Treatment on Mechanical Allodynia, Strength and Locomotor Activity

The BTZ protocol used in this study was well tolerated by mice and no overt signs of general toxicity occurred. We did not register any significant alteration in body weight among the different groups during the entire experimental protocol (data not shown).

As shown in Figure 1A, BTZ induces the development of a progressive mechanical allodynia already evident 7 days after the first BTZ injection. As illustrated in Figure 1B, at the end of the protocol (day 28), BTZ can reduce the grip strength however BTZ treatment does not compromise locomotor activity; in fact, we do not register any alteration in the t-turn, t-walk and t-total time in BTZ mice (Figure 1C).

The treatment with PKRs antagonist PC1 is able to counteract the development of allodynia (Figure 1A) and grip strength reduction (Figure 1B) due to BTZ treatment without affecting behavior in CTR mice.

### 2.2. Effect of BTZ Treatment on Mood

Figure 2 illustrates the effect of BTZ treatment on the development of anxiety (Figure 2A–C) and depressive-like behavior (Figure 2D–F). As shown in panels A and B, BTZ, at the end of the protocol, induces in the animals the development of anxiety-like behavior, evaluated by means of dark/light test and expressed as a decrease in the time spent in the light area (Figure 2A) and a decrease in the number of transitions between the light and the dark compartment (Figure 2B) in comparison to CTR mice. However, no difference in the number of marbles buried is evident among the experimental groups (Figure 2C) suggesting the absence of a compulsive behavior in BTZ mice. As shown in the lower panels of the same figure, BTZ treated mice manifest depressive-like behavior evaluated by swimming forced test. In these animals, a reduction of the first immobility time (Figure 2D) and a significant increase in the immobility time in the last 4 min of test (Figure 2E) is evident. BTZ mice do not display anhedonic symptoms; in fact, no alterations are evident in sucrose preference test between CTR and BTZ mice (Figure 2F). In BTZ mice, the treatment with PC1 can prevent the development of depressive-like behavior (Figure 2D and E) but can only partially oppose to anxiety (Figure 2A,B). PC1 does not affect any of the analyzed parameters in CTR mice.

### 2.3. Effect of BTZ Treatment on Hippocampus (HPC) Neuroinflammation

In the hippocampus of BTZ mice (Figure 3) a significant increase in PK2 and PKR2 mRNA (Figure 3A and C, respectively) is evident. In this area, a significant upregulation of the mRNA levels of the proinflammatory cytokines IL-6 (Figure 3D) and TNF-α (Figure 3E) is also present. PC1 is able to counteract PK system alterations (Figure 3A,C) and to prevent the upregulation of IL-6 (Figure 3D) without, however, showing a significant effect on TNF-α (Figure 3E). No significant changes are evident regarding PKR1 (Figure 3B), TLR4 (Figure 3F) and glial markers (Figure 3G,H). As shown in panel I, a decrease in BDNF mRNA levels is present in the hippocampus of BTZ mice and PC1 significantly contrasts it, by keeping the neurotrophin at CTR values. PC1 does not affect any of the considered parameters in CTR mice.

### 2.4. Effect of BTZ Treatment on Prefrontal Cortex (PFC) Neuroinflammation

As shown in Figure 4, in the PFC of BTZ mice a clear neuroinflammation is evident. In fact, in this tissue, an upregulation of the mRNA levels of proinflammatory cytokines, IL-6 (Figure 4D) and TNF-α (Figure 4E), of TLR4 (Figure 4F) and of the microglial marker CD11b (Figure 4G) is present. No significant alterations are observed regarding PK system (Figure 4A–C) and the astrocytic marker GFAP (Figure 4H). As shown in panel I, a significant decrease in the BDNF mRNA levels is evident in BTZ mice. The treatment with the PKRs antagonist PC1 is able to prevent the development of neuroinflammation in PFC keeping proinflammatory cytokines (Figure 4D,E), TLR4 (Figure 4F) and CD11b (Figure 4G) at control values. On the contrary, PC1 is unable to counteract the downregulation of BDNF (Figure 4I). As shown in figure, PC1 treatment does not affect any of the considered parameters in CTR mice.

### 2.5. Effect of BTZ Treatment on Hypothalamus (HPT) Neuroinflammation

A clear neuroinflammatory condition is also evident in the hypothalamus of BTZ treated mice. In fact, as shown in Figure 5, significant increases of PK2 (Figure 5A), IL-6 and TNF-α (Figure 5D,E) and of the glial markers CD11 (Figure 5G) and GFAP (Figure 5H) are present in BTZ mice. No other significant differences emerge for the other parameters analyzed. PC1, when administered to BTZ mice, is able to prevent the biochemical alterations without altering any parameter when administered in CTR mice.

### 2.6. Morphological Changes of Hippocampus and Prefrontal Cerebral Cortex Observed by Light and Transmission Electron Microscopy

Figure 6 illustrates morphological changes of hippocampus (Figure 6A) and prefrontal cortex (Figure 6B,C) in BTZ treated animals observed by light and transmission electron microscopy. Hippocampal sections from CTR mice show orderly arranged pyramidal neurons with a poorly electron-dense cytoplasm and pale nuclei containing dark nucleoli (Figure 6(Aa,c)). As shown in Figure 6(Ab,d), sections from hippocampus of BTZ treated mice present pale and normally stained neurons but also several intensely stained neurons with dark cytoplasm. No ultrastructural morphological changes in dark neurons are observed by transmission electron microscopy and cytoplasmic organelles appeared intact. This condition is similar in all hippocampal regions.

Light microscopy analysis of toluidine blue-stained coronal prefrontal cortex sections of CTR mice show normal-shaped nerve cell bodies with both classic pale cytoplasm and nuclei (Figure 6(Ba)). At the ultrastructural level neurons from CTR mice show regular endoplasmic reticulum and mitochondria with a dense matrix, provided of organized cristae (Figure 6(Bc)). In sections from BTZ-treated mice, several considerably shrunken neurons with electron-dense somata are evident, scattered among apparently intact neurons in an otherwise normal-looking environment (Figure 6(Bb)). At the ultrastructural level dark neurons exhibit some cytoplasmic organelles with signs of degeneration as dilated endoplasmic reticulum and some swollen mitochondria (Figure 6(Bd); Figure 6(Ca,b)). Around some dark nerve cell bodies, membrane-bound large empty spaces are present, probably representing marked swollen astrocytic processes (Figure 6(Cb)). Furthermore, in several blood vessels perivascular edema, corresponding to dilated feet processes of astrocytes, is clearly visible (Figure 6(Cc,d)).

## 3. Discussion

Chemotherapy has enhanced cancer patient survival, however, antitumoral drugs are often related to the development of off-target side effects. Among these, chemotherapy-induced painful neuropathy (CIPN), chemotherapy-induced cognitive impairment (CICI) and emotional distress are taking on particular importance considering their high impact on patient’s life quality [1,2,3,4,31]. The development of such effects is independent of the anticancer mechanism, it is in fact common to both classical and more recent anticancer drugs such as proteasome inhibitors, like bortezomib (BTZ) or also to more innovative anticancer strategies like monoclonal antibodies and immune checkpoint inhibitors [4,5,6,32,33]. Furthermore, the appearance of side effects represents the main cause of dose reduction or treatment discontinuation in cancer patients. Oxidative stress and neuroinflammation represent common conditions for the development of chemotherapy-induced side effects [4,6,7,10] and recent evidence indicate that side effects are commonly connected and dependent on each other [11,12,13,16]. In addition, a close relationship between chronic pain and psychiatric disorders has been recently suggested and correlated to the presence of an overlap between pain and emotional processing pathways [11,12,14,15,16,17].

As already demonstrated by us and others [28,34,35], we confirm here that BTZ treatment leads to painful neuropathy development. Hypersensitivity is precociously evident in mice; indeed, it is already present 7 days after the first BTZ administration and becomes more evident at the end of the protocol (day 28) when muscle strength reduction is also observed. In addition, we demonstrate here for the first time that, at the end of BTZ treatment, in presence of a well-established chronic pain condition, BTZ mice exhibit depressive- and anxiety-like behavior evaluated by forced swim and dark/light test, respectively. However, mice do not display a compulsive or a clear anhedonic behavior suggesting that those two particular aspects of anxiety and depression are not affected in this condition [36,37]. The differences among groups observed in the behavioral tests for evaluating mood disorders are not biased by changes in locomotor activity in fact no alterations are detected at the static rod test between BTZ and CTR mice. We and others previously [28,34,38] demonstrated that neuroinflammation plays an important role in the development of BTZ-induced neuropathic pain. The damage due to BTZ cytotoxicity in the PNS promotes the recruitment of CD68 positive cells with consequent release of immunomodulatory molecules, such as cytokines and chemokines, that are important in the sensitization process. Among chemokines, we described a crucial role of the prokineticin-PK2 in sustaining a neuroinflammatory condition that moves from the periphery to central nervous system and plays a key role in chronic pain maintenance and exacerbation [28,29].

Due to the lack of a blood–brain barrier (BBB), PNS is particularly susceptible to chemotherapeutics. However, BBB integrity could be affected by ROS release, inflammation and TLR4 activation at brain level, occurring after chemotherapy [1,7,18,19], thus allowing to peripheral inflammatory mediators or drug metabolites to reach brain areas promoting or sustaining supraspinal neuroinflammation.

Hippocampus (HPC) and prefrontal cortex (PFC) are key sites in emotional control [21], as well as regulators of the affective and emotional component of pain. Both brain areas are also involved in the regulation of the hypothalamus-pituitary-adrenal (HPA) axis [21,22], which is in turn involved in emotional control. Here, we show that anxious and depressed BTZ-treated mice are characterized by neuroinflammation in PFC, HPC and HPT. In all these brain areas we find increased mRNA expression levels of the proinflammatory cytokines IL-6 and TNF-α associated with increased mRNA levels of CD11b in PFC or CD11b and GFAP in hypothalamus. It is known that BTZ poorly passes [39,40,41] BBB and it is possible to hypothesize that the absence of a complete BBB barrier [42] makes hypothalamus more susceptible to a direct BTZ action. As far as we know, no papers in literature report increased mRNA expression of CD11b or GFAP in brain regions following BTZ. A modulation and activation of microglia or astrocytes in selected brain areas related to central pain sensitization have indeed been observed in oxaliplatin-induced neuropathic pain in rats. Since oxaliplatin, like bortezomib, poorly passes the BBB, authors suggest that these alterations are probably due to the presence of chronic pain [43,44]. However, on the basis of our data, we cannot state if the neuroinflammatory condition observed in brain regions is due to the low amount of BTZ that could eventually reach the brain, to chronic neuropathic pain or to a combination of these events.

BTZ treated mice are also characterized by low mRNA levels of the neurotrophin BDNF in HPC and PFC. Our data are in accordance with literature that describes, in the same brain regions, the presence of morphological and biochemical alterations in patients affected by chronic pain and depressed subjects. A key role of the proinflammatory cytokines IL-6 and TNF-α [20,23,24,25,26,27] has in fact been described in depression, moreover, both in patients and experimental models a clear relationship between low levels of BDNF and the presence of anxiety and depression has been suggested. BDNF has in fact an important role in the maintenance and survival of neurons and in synaptic plasticity [26,45] and an abnormal neural plasticity is often associated with depression [46]. In addition, it was recently defined a role of PK2 in mood control [30]. The intracerebroventricular injection of PK2 induced in mice the development of depressive- and anxiety-like behaviors, while mice lacking the *pk2* gene displayed significantly reduced anxiety and depression. In accordance, here we demonstrate that BTZ neuropathic mice with anxiety and depressive behavior display increased PK2 mRNA levels in HPC and HPT. The potential PK2 presence in tissues expressing its receptors represents an important regulatory mechanism. Immune and endothelial cells, neurons and glia express PKRs receptor and produce PK2 so, PK2 could act in an autocrine or paracrine way sustaining an inflammatory loop [47,48]. G-CSF but also cytokines released in injured tissues are important inducers of PK2 through a STAT 3 mechanism mediated. In particular, phosphorylated STAT 3 can directly bind to the Pk2 promoter inducing PK2 production [48]. PK2 may contribute to sustain a neuroinflammatory condition promoting in turn the production of other proinflammatory cytokines which stimulate astrocytes and neurons to induce further PK2. Moreover, it was suggested that the overexpression of PK2 and PKR2 on astrocytes represents an astrocytic autocrine growth factor [48].

Although we did not directly identify which cells (neurons, astrocyte or microglia) overexpress PK2 and its receptors, solid data are present in the literature regarding PK system distribution in CNS. It is in fact clearly demonstrated that in presence of several brain insults (hypoxia, ischemic damage, neurodegenerative diseases) PK2 and PKR2 are usually overexpressed by both neurons and astrocytes in specific hippocampal areas and by the same cells in cortical neuron cultures [49,50,51]. Moreover, as we recently described, neurons and astrocytes also represent the main PK2 source in spinal cord in experimental chronic pain conditions [52].

The light and transmission electron microscopy evaluations, supporting biochemical data, confirm the presence in BTZ treated mice of an altered morphology in the two areas mainly involved in emotional control: PFC and hippocampus. In both areas, we detect the presence of dark neurons and, in the PFC, these cells present consistent ultrastructural alterations such as the presence of cytoplasmic organelles with signs of degeneration as dilated endoplasmic reticulum and some swollen mitochondria suggesting a more compromised condition in PFC than in the hippocampus. A few studies describe dark neurons as a measure of neuronal damage; in fact, their presence is often the result of a neurotoxic, hypoxic or chronic stress condition [53,54]; furthermore, the same studies suggest that the degree of neuron impairment is decisive for the neuron fate itself (die or recover).

BBB integrity is fundamental for preventing drug entry into CNS and several papers suggest that BTZ poorly passes BBB [39,40,41]. However, our ultrastructural observations show endothelial cells morphologically intact surrounded by astrocytic processes extremely dilated/swollen so, we cannot rule out that BTZ or its metabolites could reach CNS exerting a cytotoxic effect. On the basis of our knowledge, we can assume that the observed supraspinal alterations could be the result of a neuroinflammatory process that develops inside the brain as a consequence to sensitizing stimuli incoming from PNS as well as a direct BTZ cytotoxic effect. The concept of chemotherapy-induced BBB opening was recently suggested by Branca and colleagues [18] for oxaliplatin offering an additional and novel point of view on chemotherapy-induced neurotoxicity.

In contrast to the results presented in this paper, in a previous study of our group [29], in which neuropathic pain was induced with the chemotherapeutic vincristine (VCR), we did not detect any emotional alteration in neuropathic mice. It is important to underline that the two experimental protocols are different; in the BTZ protocol, mice have been in chronic pain for 21 days, while in the VCR protocol, they have only been in chronic pain for 10 days. We can assume that 10 days of chronic pain are not enough to induce a significant neuroinflammatory activation at supraspinal level that, as previously mentioned, is often related to mood disorders. This condition is instead evident after BTZ treatment in which pain has been present for a longer time. If a correlation between the presence of mood alterations and pain duration can be confirmed, the importance of prompt pain treatment will be of even greater impact.

Here, we demonstrate that the treatment with the antagonist PC1, contrasting pain and reducing the mRNA levels of neuroinflammatory markers, can prevent the development of depression. Future investigations should be aimed at verifying if changes at transcriptional level also correspond to changes in protein content. However, in our previous studies in different models of neuropathic pain, we found a good parallelism between mRNA and protein levels of PK2 and cytokines in nervous tissues [52,55].

We previously [28] demonstrated that PC1 treatment opposed to PNS neuroinflammation, reducing the levels of CD68 and of proinflammatory cytokines that were precociously upregulated in DRG and sciatic nerve of hypersensitized BTZ mice. As a consequence, PC1 turned off a neuroinflammatory loop, also sustained by PK system activation, and prevented spinal cord sensitization resulting in pain reduction. We can suppose that, blocking this circuit, the antagonist blocks the neuroinflammatory pathway that from PNS reaches brain areas and plays a role in priming or sustaining neuroinflammation at supraspinal level. On the other hand, considering that PC1 could pass the BBB [52], we cannot rule out the possibility that PC1 may also directly act on PKRs expressed in the brain [47,48]. In view of the key role of cytokines and neuroinflammation in mood disorders and pain and considering the pivotal role of PKs in neuroinflammation [23,27,48], we can suppose that the PKRs block could turn off a neuroinflammatory condition responsible of the pathological state.

However, our data show that although PC1 has a strong ability to normalize altered mRNA levels of most neuroinflammatory markers evaluated it was unable to significantly oppose to TNF-α increase in the hippocampus. Considering the crucial role of this cytokine in anxiety [56,57], we can speculate that the high hippocampal TNF-α levels can contribute to sustain an anxiety behavior in BTZ mice.

We are aware that our experiments, as often happens when chemotherapy side effects are investigated in experimental models, were performed in naïve animals and not in tumor-bearing mice, a condition that could in turn impact on the emotional component. Interestingly, PKs have been shown to have a role in tumor progression inducing angiogenesis and sustaining inflammation [58,59]. In particular, PK1 or EG-VEGF is important in the neo vascularization process and is involved in multiple myeloma cells proliferation and survival [60] while PK2 is a recognized proinflammatory cytokine [58]. Moreover, it was demonstrated that the treatment with anti-PK2 antibodies was effective in reducing tumor growth and angiogenesis and exerted an additive antitumor effect in combination with anti-Vegf agents or cisplatin [61]. The evidence would, therefore, suggest a possible broader beneficial effect of PKRs antagonism; however, before thinking of a possible use of PC1 in cancer patients, the possible effect of the antagonist in mice with tumor should be tested.

In conclusion, our study suggests that the therapeutic treatment with the PKRs antagonist PC1, contrasting neuroinflammation, can counteract or prevent the main complications of chemotherapeutics such as neuropathic pain and mood alterations like depression.

## 4. Materials and Methods

### 4.1. Animals

Nine-week-old C57BL/6J male mice (Charles River Laboratories, Calco, Italy), were used. Animals were acclimatized for 1 week before starting experiments with 12 h dark/light cycle at 22 °C ± 1 °C room temperature and humidity of 55% ± 10% and food and water ad libitum. During acclimatization period, mice were handled by exposure to a passive hand, tickling and hand restraint for few minutes/day.

All animal experiments comply with ARRIVE guidelines and were carried out in accordance with EU directive 2010/63/EU for animal experiments and International Association for the Study of Pain and European Community (E.C.L358/118/12/86) guidelines, and were approved by the Animal Care and Use Committee of the Italian Ministry of Health (permission number 709/2016 to SF; approval date 22 July 2016). All efforts were made to minimize suffering and the number of animals used, in accordance with 3R principles.

### 4.2. Neuropathy Induction and Therapeutic Treatment with PC1

Bortezomib (BTZ, LC Laboratories; Woburn, MA, USA) was dissolved at 1 mg/mL in dimethyl sulfoxide (DMSO; SigmaAldrich, Milan, Italy) and diluted in sterile 0.9% NaCl (saline) solution to a final concentration of 40 μg/mL. BTZ or vehicle was then intraperitoneally (i.p.) injected at the dose of 0.4 mg/kg three times a week (every Monday, Wednesday, Friday) for a total of four consecutive weeks. After confirming the presence of hypersensitivity, the PKRs antagonist PC1 [28,62], a triazine-guanidine compound, was dissolved in sterile saline and subcutaneously administered to BTZ mice (BTZ + PC1) or to naïve mice (PC1) at the dose of 150 μg/kg twice a day for 14 consecutive days, from day 14 until the end of the BTZ schedule (day 28) [28]. The healthy status of animals and the presence of any toxicity sign was assessed during the entire experimental protocol (body weight, coat, posture, food and water intake, grooming, stools).

### 4.3. Behavioral Tests

Mice were randomly distributed in two different cohorts to evaluate, at the end of the BTZ treatment (day 28), the presence of anxiety- or depressive-like behavior. Moreover, before (0) and after 7, 14, 21 and 28 days of BTZ, half animals for each cohort performed motor/locomotor tests (strength, balance and coordination) and the other half pain-like behavioral test (mechanical allodynia). Mice were acclimatized in the testing room for 30 min before any evaluation. Behavioral tests were performed, in standardized conditions, by blinded trained experimenters.

### 4.4. Von Frey Test: Mechanical Allodynia

Mechanical allodynia was evaluated through mechanical touch sensitivity using a blunt probe (Von Frey filament, 0.5 mm diameter, ranging up to 10 g in 10 s) on central plantar surface of hind-paw by dynamic plantar aesthesiometer (UgoBasile, Gemonio, Italy). Responses to mechanical stimuli (Paw Withdrawal Thresholds, PWT) were expresses in grams (g) [29]. Three different measurements for each paw were recorded and the mean calculated.

### 4.5. Static Rod Test: Coordination and Balance

A 60 cm long wooden rod (Ø 22 mm), 80 cm elevated above the padded floor, was fixed to a laboratory bench so that, the rod horizontally protrudes into the space. Rod-end near the counter was the arrival. The mouse was placed at the end of central plantar surface rod, opposite to the counter, with the nose oriented towards the space. The orientation time (*t-turn*, time spent to orientate 180° from the starting position to the bench) and transit time (*t-walk*, the time taken to travel to the bench) were recorded. The sum of these two parameters represents the total time for test’s execution (*t-total*). If, during orientation, (*t-turn*) the mouse flips over and/or clings below the rod, falls or fails to orient the experimenters assign it the maximum time (120 s, cut-off) for both t-turn and t-walk. If, during the transit (*t-walk*), the mouse falls off the rod, the experimenters assign it the maximum score of 120 s [63]. Before performing the test, mice were trained to complete all test phases.

### 4.6. Grip Strength Test: Muscle Force

The grip strength test (or wire test) was used to assess mice force. The apparatus consists of a 2 mm diameter metal wire laterally supported by two supports and maintained at 60 cm altitude. Mice were allowed to cling with the only anterior paws and during the test the experimenter registers the time the mice fall before reaching the lateral supports or the time till one of the forepaws touched the side support. Maximum test time (cut-off time) is 30 s. A score (range 1–6), was assigned to each mouse according the following criteria: falling between 1 and 5 s = 1; falling between 6 and 10 s = 2; falling between 11 and 20 s = 3; falling between 21 and 30 s = 4; no falling in 30 s = 5; touching side support without falling = 6 [63].

### 4.7. Dark Light Test: Anxiety-Like Behavior

The apparatus consists of two-chambers arena: a black chamber (one third of the total area) with black walls and lid, and a white uncovered chamber with white walls and exposed to light. An opening located at the center bottom of the divider allows mice to move towards the two chambers. Each mouse was placed in the white chamber and allowed free to explore the two-chambers arena for 5 min. The time spent in the white chamber and number of transitions between the chambers were registered [29].

### 4.8. Marble Burying Test: Anxiety-Like Behavior

Cages were filled with about 5 cm of sawdust bedding and 18 marbles were placed on the floor in a regular pattern. The mouse was then placed into the cage and left undisturbed for 30 min. At the end of the session, the number of unburied marbles (>1/3 outside the sawdust) was counted [64].

### 4.9. Swimming Forced Test: Depression-like Behavior

Apparatus consists of a 3 L glass beaker filled with water (23–25 °C). The mouse was gently put into the beaker and allowed to swim undisturbed for 6 min. At the end of the session, the animal was removed, wiped up, and replaced to its home cage. Water was changed between each animal. During the test, the latency time, corresponding to the first time of lasting immobility (>10 s) and the immobility time during the last 4 min of session, were recorded [29].

### 4.10. Sucrose Preference Test: Anhedonia/Depression-like Behavior

Prior to beginning test, each mouse was exposed to the presence of two drinking bottles (both containing water) for 48 h. Then, following this acclimation period, each mouse was exposed for 24 h to one bottle of 2% sucrose solution (Sucrose 99%, AlfaAesar; Karlsruhe, Germany) and one of regular water, both weighed before and after the test and switched in the position to reduce any confound produced by a side bias. Sucrose preference was calculated as a percentage of the volume of sucrose intake over the total volume of fluid intake for each tested [29] animal.

### 4.11. Brain Areas Collection and Real Time-qPCR

At the end of the BTZ schedule (day 28) corresponding to 14 days of PC1 treatment, mice were sacrificed by decapitation. Brain was excised and hippocampus (HPC), prefrontal cortex (PFC) and hypothalamus (HPT) were collected from each animal. RNA was extracted from homogenized tissue using TRIzol^®^ reagent (Invitrogen, Carlsbad, CA, USA) according to manufacturer’s instruction. cDNA was obtained from 1000 ng RNA using reverse transcriptase kit LunaScript™ (BioLabs; London, UK). mRNA levels of: prokineticin 2 (PK2; Mm01182450_g1), prokineticin receptors (PKR1 and PKR2; Mm00517546_m1, Mm00769571_m1), interleukin 6 (IL-6, Mm00446190_m1), tumor necrosis factor alpha (TNFα, Mm00443258_m1), Toll-like receptor 4 (TLR4, Mm00445274_m1), Cluster Of Differentiation 11b (CD11b, Mm00434455_m1), glial fibrillary acidic protein (GFAP, Mm01253033_m1), and brain derived neurotrophic factor (BDNF, Mm 04230607_s1) were measured with quantitative PCR by QuantStudio 5™, (Thermofisher Scientific, Waltham, MA, USA) using Taqman Gene expression assays and Luna^®^ Universal Probe qPCR Master Mix (BioLabs; London, UK). The mRNA levels were normalized to GAPDH (Mm99999915_g1) and expressed as 2^−ΛΛCT^ values relative to control group. Each sample was run in triplicates alongside non-template controls [29].

### 4.12. Light and Transmission Eelectron Microscopy

At day 28, CTR and BTZ treated mice were deeply anaesthetized with 60 mg/Kg Zoletil (Tiletamine/Zolazepam) and transcardially perfused with a solution containing 2% formaldehyde and 2% glutaraldehyde in 0.1 M sodium cacodylate buffer (pH 7.3). Brain was quickly removed from each animal and post-fixed for an additional 24 h at 4 °C. Coronal slices of 100 μm thickness were cut by using a Leica VT1000S Vibratome. Sections were collected in 0.1 M sodium cacodylate buffer, and hippocampus and prefrontal cerebral cortex were manually dissected. Subsequently, samples were washed in cacodylate buffer and postfixed at 0 °C for 90 min in 2% osmium tetroxide (Sigma-Aldrich) in the same buffer, washed in distilled water and stained in 2% aqueous uranyl acetate. Then, it was carried out the dehydration in ethyl alcohol and the embedding in Epon-Araldite resin. Semithin sections (0.5 μm thick) of each sample were stained with 0.5% toluidine blue in 1% sodium borate and examined with a light microscope (Zeiss Axiophot). Ultrathin sections (50 to 70 nm thick), cut on a Leica Supernova ultramicrotome, were counterstained with lead citrate and examined under a Zeiss EM10 electron microscope (Oberkochen, Germany) [28].

### 4.13. Statistical Analysis

Data were expressed as mean ± SEM (8 animals/group for behavioral and biochemical evaluations). Results obtained from pain-related behavioral and locomotor activity were analyzed using two way-ANOVA analysis of variance with repeated measures followed by Bonferroni’s test for multiple comparisons. Anxiety- and depressive-like behaviors results and biochemical evaluations, performed at day 28, were analyzed by one way-ANOVA followed by Bonferroni’s test. Statistical analysis was performed blinded using GraphPad 6 (San Diego, CA, USA). Differences were considered significant at *p* ≤ 0.05.

## Figures and Tables

**Figure 1 ijms-22-10256-f001:**
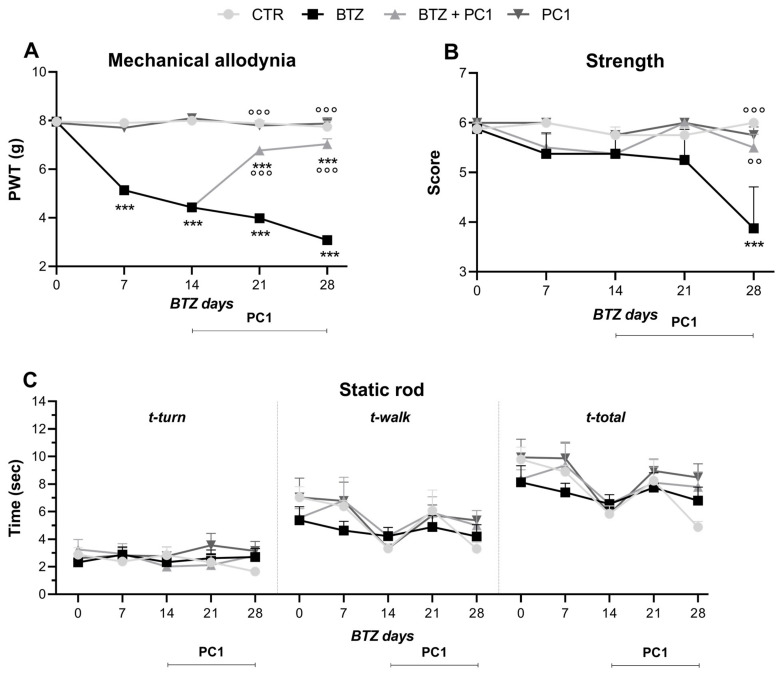
Mechanical allodynia, strength and locomotor evaluations. Rating of mechanical allodynia evaluated by dynamic plantar aestesiomether (**A**), strength evaluated using wire test score (**B**) and motor coordination and balance (*t-turn*, orientation time; *t-walk*, travel time; *t-total*, total time) evaluated by static rod test (**C**). BTZ was intraperitoneally (i.p.) administered from day 0 to day 28 (0.4 mg/kg, 3 times week/4 weeks); PC1 was subcutaneously (s.c.) administered (150 μg/kg, twice daily) from day 14 until day 28. Data are presented as the mean ± SEM of 8 mice/group. Statistical analysis was performed by means of Two-way ANOVA for repeated measures followed by the Bonferroni’s post-test. *** *p* < 0.001 vs. CTR; °° *p* < 0.01, °°° *p* < 0.001 vs. BTZ.

**Figure 2 ijms-22-10256-f002:**
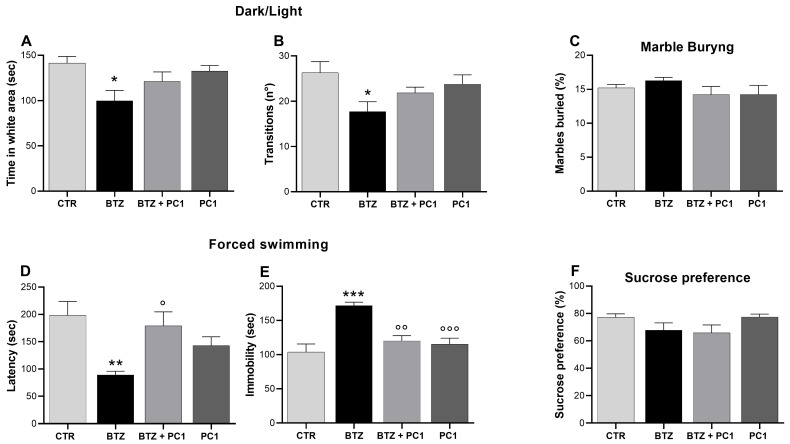
Anxiety- and depressive-like behaviors. Mood behavioral evaluations were performed at the end of the experimental protocol at day 28 (BTZ i.p. administered from day 0 to day 28, 0.4 mg/kg, 3 times week/4 weeks; PC1, s.c. administered, 150 μg/kg, twice daily, from day 14 until day 28) in two different cohorts of animals. Anxiety-like behavior was assessed by means of dark/light test, evaluating the time spent in the white area (**A**) and number of transitions between the white and dark area (**B**), and by means of marble burying test (**C**). At the same time, day 28, depressive-like behavior was tested using the forced swimming test, evaluating both the latency, corresponding to the first stop time (**D**), and the immobility time in the last 4 min of the test (**E**), and by means of sucrose preference test (**F**). Data are presented as mean ± SEM of 8 animals per group. Statistical analysis was performed by One-way ANOVA analysis of variance followed by Bonferroni’s post-test. * *p* < 0.05, ** *p* < 0.01, *** *p* < 0.001 vs. CTR; ° *p* < 0.05, °° *p* < 0.01, °°° *p* < 0.001 vs. BTZ.

**Figure 3 ijms-22-10256-f003:**
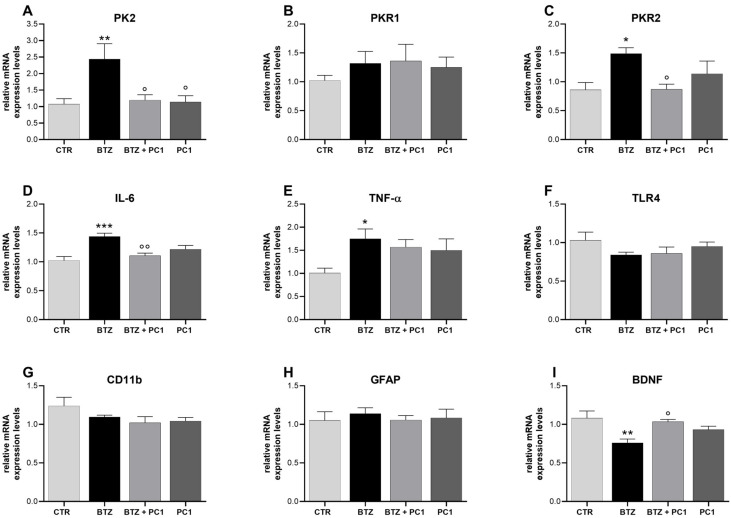
Biochemical evaluations in the hippocampus. mRNA levels of prokineticin system PK2, PKR1, PKR2 (**A**–**C**), pro-inflammatory cytokines IL-6 (**D**) and TNF-α (**E**), TLR4 (**F**), glial markers CD11 and GFAP (**G**,**H**), and of the neurotrophin BDNF (**I**) were measured in hippocampus by means of Real Time-qPCR. Evaluations were performed at the end of BTZ protocol, 28 days after the first BTZ administration (BTZ i.p., 0.4 mg/kg, 3 times week/4 weeks) and after PC1 administration (150 μg/kg, s.c., twice daily, from day 14 until day 28). Results were expressed in relation to GAPDH and presented as fold-increases over the levels of CTR-mice. Data are expressed as the mean ± SEM from 8 mice/group. Statistical analysis was performed using One-way ANOVA followed by Bonferroni’s post-test. * *p* < 0.05, ** *p* < 0.01, *** *p* < 0.001 vs. CTR; ° *p* < 0.05, °° *p* < 0.01 vs. BTZ.

**Figure 4 ijms-22-10256-f004:**
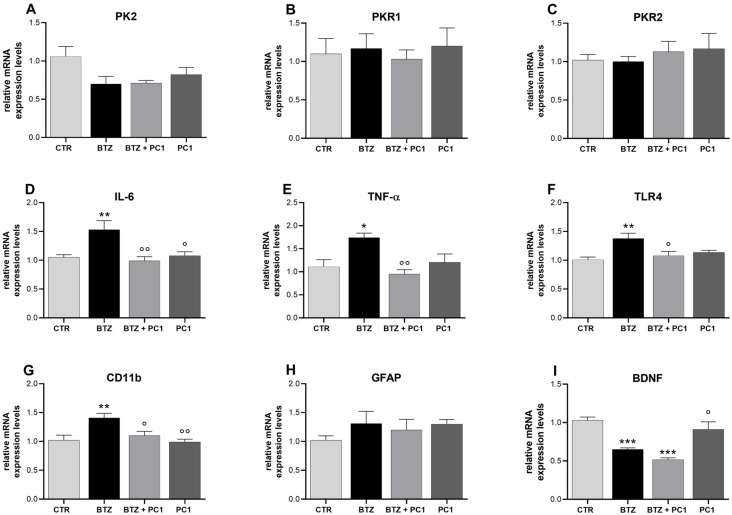
Biochemical evaluations in the prefrontal cortex. mRNA levels of prokineticin system PK2, PKR1, PKR2 (**A**–**C**), pro-inflammatory cytokines IL-6 (**D**) and TNF-α (**E**), TLR4 (**F**), glial markers CD11 and GFAP (**G**,**H**), and of the neurotrophin BDNF (**I**) were measured in the prefrontal cortex by means of Real Time-qPCR. Evaluations were performed at the end of BTZ protocol, 28 days post the first BTZ administration (BTZ i.p., 0.4 mg/kg, 3 times week/4 weeks) and after PC1 administration (150 μg/kg, s.c., twice daily, from day 14 until day 28). Results were expressed in relation to GAPDH and presented as fold-increases over the levels of CTR-mice. Data are expressed as the mean ± SEM from 8 mice/group. Statistical analysis was performed using One-way ANOVA followed by Bonferroni’s post-test. * *p* < 0.05, ** *p* < 0.01, *** *p* < 0.001 vs. CTR; ° *p* < 0.05, °° *p* < 0.01 vs. BTZ.

**Figure 5 ijms-22-10256-f005:**
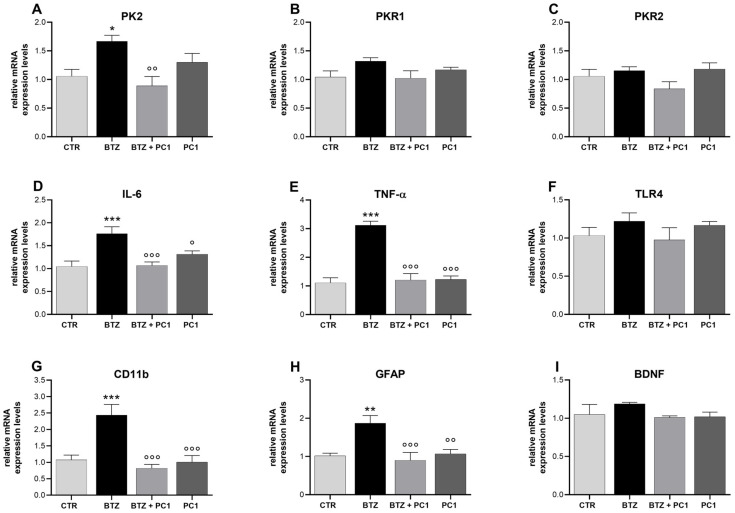
Biochemical evaluations in the hypothalamus. mRNA levels of prokineticin system PK2, PKR1, PKR2 (**A**–**C**), pro-inflammatory cytokines IL-6 (**D**) and TNF-α (**E**), TLR4 (**F**), glial markers CD11 and GFAP (**G**,**H**), and of the neurotrophin BDNF (**I**) were measured in the hypothalamus by means of Real Time-qPCR. Evaluations were performed at the end of BTZ protocol, 28 days post the first BTZ administration (BTZ i.p., 0.4 mg/kg, 3 times week/4 weeks) and after PC1 administration (150 μg/kg, s.c., twice daily, from day 14 until day 28). Results were expressed in relation to GAPDH and presented as fold-increases over the levels of CTR-mice. Data are expressed as the mean ± SEM from 8 mice/group. Statistical analysis was performed using One-way ANOVA followed by Bonferroni’s post-test. * *p* < 0.05, ** *p* < 0.01, *** *p* < 0.001 vs. CTR; ° *p* < 0.05, °° *p* < 0.01, °°° *p* < 0.001 vs. BTZ.

**Figure 6 ijms-22-10256-f006:**
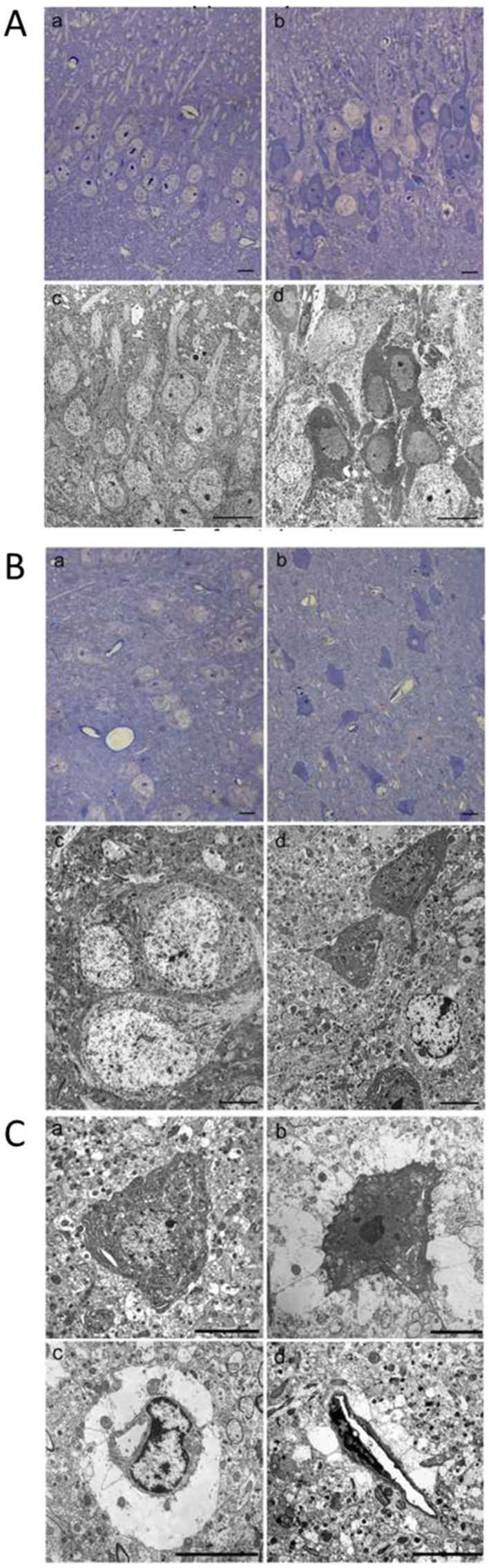
Light and electron microscopy images of hippocampus and prefrontal cortex. (**A**) light (**a**,**b**) and electron (**c**,**d**) microscopy images of neuronal cell bodies of hippocampus isolated from CTR or BTZ mice. In control mouse neurons (**a**,**c**) are orderly arranged and cytoplasm appear clear with pale nuclei. At the end of BTZ treatment (day 28), several nerve cell bodies (**b**,**d**) exhibit a dark cytoplasm. Scale bars: 10 µM. (**B**) light (**a**,**b**) and electron (**c**,**d**) microscopy images of neuronal cell bodies of prefrontal cerebral cortex isolated from CTR or BTZ mice. Neurons of CTR mice (**a**,**c**) show normal shape with clear cytoplasm and pale nuclei. After treatment with BTZ (day 28) some nerve cell bodies (**b**,**d**) appear shrunk and exhibit a dark cytoplasm. Scale bars: 10 µM. (**C**) shows representative transmission electron microscopy images exhibiting some details from prefrontal cerebral cortex at the end of BTZ treatment (day 28). (**a**,**b**) dark shrunk neurons. (**a**) a nerve cell body containing dilated endoplasmic reticulum. (**b**) a dark neuron contains a dark nucleus with electron dense nucleolus and swollen mitochondria; just around this neuron, large empty spaces are present. (**c**,**d**) perivascular edema, corresponding to dilated feet processes of astrocytes, are clearly visible around the blood vessels. Scale bars: 5 µM.

## Data Availability

The datasets used and/or analyzed during the current study are available from the corresponding author on reasonable request.

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
