# Peer review of "The Antagonism of the Prokineticin System Counteracts Bortezomib Induced Side Effects: Focus on Mood Alterations"

_ijms, 2021, doi:10.3390/ijms221910256_

Round 1

Reviewer 1 Report

The authors show the crucial relationship between BTZ-induced mood alterations and PK2-PKR2 signaling. The study design is entirely good, and there is a certain novelty. However, I recommend the authors to add some discussions and to do immunohistostaining experiments to corroborate their data interpretation.

If the authors claim that their present data firstly demonstrates that BTZ treatment activates glial cells in the brain, they should correctly evaluate glial morphology and cell numbers in the IHC experiments. I think this additional IHC experiment must not be difficult. Furthermore, by using the same samples for evaluating glial activation on the IHC, the authors can try to observe which cell type (neuron, astrocyte, or etc.) are increased the PK2 and PKR2 expression after BTZ treatment, if there are available good antibodies.

These additional experiments will enable us to better understand about whether the increases of mRNA expression of PK2 systems reflect the increase of these genes-expressing cell number, or the increase of gene expression in a cell.

I think that Fig 3A and Fig. 5A suggest that there is the presence of feed-forward loop of PK2 production (PK2 induces additional PK2). Are there any information about the feed-forward loop of PK2 in the physiological and/or pathological conditions? I recommend the authors to add discussion about that.

Reviewer 2 Report

The article is interesting, the topic is actual. The article is devoted to the mechanisms of the development of side effects of chemotherapy treatment, for example, induced painful neuropathy, cognitive impairment caused by chemotherapy and psychological stress, anxiety and depression. These phenomena significantly affect the quality of life of patients. The article also discusses the involvement of neuroinflammation in these effects and how to prevent their development. The authors suggest that therapeutic treatment with a PKR PC1 antagonist contrasting with neuroinflammation may counteract or prevent major complications of chemotherapeutic agents such as neuropathic pain and mood changes such as depression. Thus, this article has both scientific and practical significance.

The article is well written, read with interest.

  1. What is the reason for the highlighting in yellow (lines 267-269, 298-305, 357-358, 366-369)?
  2. Line 274 is an unreasonable line break.

Reviewer 3 Report

The manuscript under review is exceptional in content and well written. Inclusion of the behavioral, as well as the major RT-PCR and light and electron microscopy changes in the corresponding brain regions strongly supports the claims that mood alterations are related to the neuroinflammation, glial biomarker change, and PK system activation in prefrontal cortex, hippocampus and hypothalamus. As final evidence the description of pygnotic changes in neurons and vasculature in these regions is offered. Other information provided is animal study certifications and descriptive information about general animal health after this chemotherapy regimen.

For better English accuracy and to oppose autocorrect, please edit the following:

In lines 152, 214, 215,  Please use the adjective form “shrunken”, rather than the verb past tense.

In line 431, Use infinitive verb, for “Mice were allowed to cling with only the anterior paws and during the test the time……”

Line 465 Add the “l” for  “Real-time PCR”

Line 474 “Toll-like receptor”

Round 2

Reviewer 1 Report

The paper has been improved in this revision.

Author Response

We thank the reviewer for appreciating the changes made to our work and for considering now our work suitable for publication 

This manuscript is a resubmission of an earlier submission. The following is a list of the peer review reports and author responses from that submission.

Round 1

Reviewer 1 Report

This manuscript is about chemotherapy, specially BTZ, related side effects, neuropathy, cognitive impairment, and psychological distress, and their prevention with a PKR antagonist, PC1. Overall, this is well written and can deliver a take-home message to the readers. Therefore, the reviewer recommends this manuscript be published in the International Journal of Molecular Sciences after minor revisions.

  1. The reviewer suggests revising the title (incomplete two sentences).
  2. Lines 80 and 235: “after the first BTZ injection” to “after 1-day BTZ injection” since the injection happened three times a day.
  3. Line 81: “BTZ treatment does not compromise…” should be “ PC1 treatment does not compromise…”
  4. Line 280: “pk2” to “PK2”

Author Response

This manuscript is about chemotherapy, specially BTZ, related side effects, neuropathy, cognitive impairment, and psychological distress, and their prevention with a PKR antagonist, PC1. Overall, this is well written and can deliver a take-home message to the readers. Therefore, the reviewer recommends this manuscript be published in the International Journal of Molecular Sciences after minor revisions.

à We thank the reviewer for appreciating our paper

The reviewer suggests revising the title (incomplete two sentences).

As suggested we have modified the title of the paper

Lines 80 and 235: “after the first BTZ injection” to “after 1-day BTZ injection” since the injection happened three times a day.

As described in materials and methods paragraph BTZ was intraperitoneally (i.p.) injected at the dose of 0.4 mg/kg three times a week (every Monday, Wednesday, Friday) for a total of four consecutive weeks. We therefore maintain in the text the expression “after the first BTZ injection”

Line 81: “BTZ treatment does not compromise…” should be “PC1 treatment does not compromise…”

In this specific point we referred to the capacity of BTZ treatment to reduce grip strength without compromising locomotor activity. 

Line 280: “pk2” to “PK2”

 In this point we referred to pk2 gene. We change pk2 in pk2 (line 66, 321)

Reviewer 2 Report

This study shows the important evidence which a painful BTZ mice model represent anxiety and depression-like behaviors. In addition, the authors show PK2-PKR2 signaling is a potential target to prevent these behaviors. The study design is entirely good, and there is a certain novelty. However, the authors should revise some points of manuscript and add several experiments to clarify the data interpretation.

- BTZ treatment induces the decrease of grip strength. Were there any general toxicities, such as decrease of body weight?

- There is a concern about anti-tumor effect of bortezomib with PC1 co-treatment. The authors should show a data or references about the effect of PC1 on anti-tumor activity of BTZ.

- It is uncertain which cell type express PK2 and PKR2 in the brain. This is highly important to interpret their data and how BTZ induces neuronal damages in hippocampus and prefrontal cortex (direct or indirect action of PK2-PKR2?).

- It is better to do the immunohistochemistry experiment to correctly evaluate glial morphological activation than mRNA levels, which should make this paper more valuable and reliable. There are many antibodies to excellently stain microglia and astrocytes. Perhaps, does this paper demonstrate the first observation which some glial cells are activated by bortezomib in several brain regions?

- I wonder if PC1 injected into periphery (s.c.) could pass the BBB. If it could not pass, the results of this study suggest that it is important to attenuate peripheral inflammation to prevent mood alteration after BTZ injection. I have several concerns about data interpretation, which PK2-PKR2 signal is critical to develop BTZ-induced anxiety and depression in peripheral or brain region (in situ)?

- It is better to represent consistently as Figure 1A, etc., throughout the manuscript, not “panel A”.

Author Response

This study shows the important evidence which a painful BTZ mice model represent anxiety and depression-like behaviors. In addition, the authors show PK2-PKR2 signaling is a potential target to prevent these behaviors. The study design is entirely good, and there is a certain novelty. However, the authors should revise some points of manuscript and add several experiments to clarify the data interpretation.

 We thank the reviewer for appreciating the novelty of our paper and for raising important points. According to reviewer’s requests we have clarified and discussed in detail the suggested points.

- BTZ treatment induces the decrease of grip strength. Were there any general toxicities, such as decrease of body weight?

We thank the reviewer to give us the possibility to clarify this point. As already observed in our previous study (Moschetti et al., 2019a) and as suggested by Bohemerle and colleagues, the BTZ protocol used in the present study is well tolerated by mice (Bohemerle et al, 2014), animals showed normal locomotor activity, social interactions and grooming behaviour without apparent distress signs moreover no animals died following BTZ treatment. We have also monitored the body weight of BTZ treated mice and we did not register any significant alteration in comparison to control mice. 

This aspect was added in the text: lines: 79-81, 423-425

“The BTZ protocol used in this study was well tolerated by mice and no overt signs of general toxicity occurred. We did not register any significant alteration in body weight among the different groups during the entire experimental protocol (data not shown).”

“The healthy status of animals and the presence of any toxicity sign was assessed during the entire experimental protocol (body weight, coat, posture, food and water intake, grooming, stools)”.

- There is a concern about anti-tumor effect of bortezomib with PC1 co-treatment. The authors should show a data or references about the effect of PC1 on anti-tumor activity of BTZ.

We thank the reviewer for raising this important point which, however, it is not the subject of investigation by this paper. It is recognized that prokineticins PKs may have a role in tumor progression (Monnier and Samson 2010; Corlan et al., 2016; Negri and Ferrara 2018). PROK1 or EG-VEGF has an important role in neo-vascularization and Li and colleagues recently demonstrated a role of the same PK1 in multiple myeloma cells proliferation and survival (Li et al., 2010); on the other hand, PK2 or Bv8 is an important pro-inflammatory cytokine. PKs have therefore the ability to directly regulate two key processes of tumoral growth both inducing angiogenesis and recruiting inflammatory immune cells, which in turn cause a positive feed-back loop and amplify tissue remodelling during inflammation (Monnier and Samson 2010). On the basis of these considerations we can suppose that the block of prokineticin receptors (PKRs) by means of PC1 can potentiate the antitumoral effect of BTZ (mainly used for multiple myeloma treatment) by blocking the over described effects of PKs.  This aspect deserves specific in-depth investigations however, in support of our considerations Ferrara’ s group described that anti-Bv8 antibodies inhibited the growth of several tumours in mice and suppressed angiogenesis. Authors also suggested that the effects of anti-Bv8 antibodies were additive to those of anti-Vegf antibodies or to cisplatin antitumor effect (Shojaei et al., 2017).

We added this point in the text, lines: 385-395

“We are aware that our experiments were performed in naïve animals and not in tumor bearing mice, a condition that could in turn impact on the emotional component. Moreover, before thinking of a possible use of PC1 in cancer patients, the possible effect of the antagonist in mice with tumor should be tested.  Interestingly, PKs have been shown to have a role in tumor progression inducing angiogenesis and sustaining inflammation [57, 58]. In particular, PK1 or EG-VEGF is important in neo vascularization process and is involved in multiple myeloma cells proliferation and survival [59] while PK2 or Bv8 is a recognized proinflammatory cytokine [57]. In addition, it was demonstrated that the treatment with anti- PK2 antibodies was effective in reducing tumoral growth and angiogenesis and exerted an additive antitumor effect in combination with anti-Vegf or cisplatin [60]”.

- It is uncertain which cell type express PK2 and PKR2 in the brain. This is highly important to interpret their data and how BTZ induces neuronal damages in hippocampus and prefrontal cortex (direct or indirect action of PK2-PKR2?).

The reviewer highlights an important point. In the present paper we only evaluated changes in the expression levels of PK members without investigating the cell type involved. However, in vitro and in vivo studies documented that in presence of several brain insults (hypoxia, ischemic damage, neuro-degenerative disease) PK2 and PKR2 are usually overexpressed by neurons and astrocytes of specific hippocampal areas and by the same cells in cortical neurons culture. (Cheng et al., 2012; Maftei et al., 2019; Zuena et al., 2019).

We have added and discussed this aspect in the discussion paragraph, lines: 327-333

“We cannot identify which cells (neurons, astrocyte or microglia) overexpress PK2 and its receptors. However, in vitro and in vivo studies present in literature documented that in presence of several brain insults (hypoxia, ischemic damage, neurodegenerative disease) PK2 and PKR2 are usually overexpressed both by neurons and astrocytes in specific hippocampal areas and by the same cells in cortical neurons culture [49- 51]. Neurons and astrocytes have been clearly demonstrated to be the main source of PK2 also in spinal cord in different experimental chronic pain conditions [52].”

- It is better to do the immunohistochemistry experiment to correctly evaluate glial morphological activation than mRNA levels, which should make this paper more valuable and reliable. There are many antibodies to excellently stain microglia and astrocytes. Perhaps, does this paper demonstrate the first observation which some glial cells are activated by bortezomib in several brain regions?

We agree with the reviewer about the presence in commerce of good antibodies for testing glial morphological activation and we have also used them in our previous paper (Moschetti et al., 2019a). Unfortunately, we did not schedule these experiments and, at this moment, we do not have samples to perform immunofluorescence experiments.  However, literature often described high correspondence between high levels of glial marker mRNA (GFAP and CD11b) and morphological activation of the same cells (Raghavendra et al., 2004). On this basis, we can assume that glial markers mRNA increase observed in brain regions of BTZ treated neuropathic mice suggests the presence of activated glial cells. To our knowledge, no other paper in literature describes glial cells activation following BTZ treatment however, modulation and activation of microglia or astrocytes in selected brain areas related to central pain sensitization have been observed in oxaliplatin-induced neuropathic pain rats. Since oxaliplatin, like bortezomib, poorly passes the BBB, authors suggest that these alterations are probably due to the presence of chronic pain (Di Cesare Mannelli et al., 2013, 2014). On the basis of the data here reported and as already discussed in the text we cannot reach a conclusion whwther the alterations observed in brain regions are due to BTZ, pain or both.

We better discussed this point in the discussion, lines: 301-309

“As far as we know, no papers in literature report glial alterations in brain regions following BTZ. A modulation and activation of microglia or astrocytes in selected brain areas related to central pain sensitization have indeed been observed in oxaliplatin-induced neuropathic pain in rats. Since oxaliplatin, like bortezomib, poorly passes the BBB, authors suggest that these alterations are probably due to the presence of chronic pain [43-44]. However, on the basis of our data, we cannot state if the neuroinflammatory modifications observed in brain regions are due to the low amount of BTZ that could eventually reach the brain, to chronic neuropathic pain or to a com-bination of these events.”

- I wonder if PC1 injected into periphery (s.c.) could pass the BBB. If it could not pass, the results of this study suggest that it is important to attenuate peripheral inflammation to prevent mood alteration after BTZ injection. I have several concerns about data interpretation, which PK2-PKR2 signal is critical to develop BTZ-induced anxiety and depression in peripheral or brain region (in situ)?

We thank the reviewer for raising this important point. It was suggested that peripherally administered PC1, at the dosage that we used, can pass BBB (Maftei et al., 2014). On the basis of these data we can assume that PC1 could exert its preventive effect on mood alteration both attenuating peripheral inflammation and counteracting central neuroinflammation acting directly on PKRs in the brain (Cheng et al., 2006; Negri and Ferrara 2018). Peripheral inflammation and neuroinflammation play an important role in mood disorders and PKs are emerging as key molecules in the neuroinflammatory process. Blocking PKR, we counteract neuroinflammation preventing the development of a pathological state

We better specify this aspect in the discussion, lines: 374-379

“On the other hand, considering that PC1 could pass the BBB [52] we cannot rule out the possibility that PC1 may counteract brain neuroinflammation also directly acting on PKRs expressed in the brain [47, 48]. In view of the key role of cytokines and neuroinflammation in mood disorders and pain and considering the pivotal role of PKs in neuroinflammation [23, 27, 48], we can suppose that the PKRs block could turn off a neuroinflammatory condition responsible of the pathological state.”

- It is better to represent consistently as Figure 1A, etc., throughout the manuscript, not “panel A”.

We have changed the text according to reviewer’s suggestions

Round 2

Reviewer 2 Report

The revised version of manuscript is made some improvement, especially in discussion section.

Actually, the authors have demonstrated that PK2-PKR2 signaling is critical for BTZ-induced mood alterations by PC1 treatment. However, a large part of their conclusions is dependent on reference papers, not from the data of this manuscript. The authors must do additional experiment by themselves.